# Future Perspectives and Conclusions from Animal Models of CHI3L1-Related Inflammation-Associated Cancer

**DOI:** 10.3390/cells14130982

**Published:** 2025-06-26

**Authors:** Emiko Mizoguchi, Siyuan Wang

**Affiliations:** 1Department of Cell Biology, Kurume University School of Medicine, Fukuoka 8300011, Japan; 2Department of Molecular Microbiology and Immunology, Brown University Alpert Medical School, Providence, RI 02912, USA; 3Department of Cell Biology, Institute of Life Science, Kurume University, Fukuoka 8300011, Japan

**Keywords:** chitinase 3-like 1, dysplasia, colitis, fibrosis, carcinogenesis

## Abstract

Among the molecules implicated in inflammation-associated tumorigenesis, Chitinase 3-like 1 (CHI3L1/YKL-40/Brp-39) has emerged as a particularly compelling target due to its multifaced roles in immune regulation, tissue remodeling, and cancer progression. Elevated CHI3L1 expression is observed in various human cancers and corresponding animal models. CHI3L1 directly promotes tumor cell proliferation and angiogenesis and also contributes to immune evasion by establishing an immunosuppressive environment in inflamed tissues. Mechanistically, CHI3L1 exerts its effects through the modulation of STAT3, MAPK, and PI3K/Akt signaling pathways and by interacting with cell surface receptors, such as IL-13Rα2 and RAGE. Studies using transgenic and knockout mouse models have revealed a strong association between CHI3L1 expression and cancer progression. In models of colon and lung cancer, CHI3L1 overexpression correlates with increased tumor size and number, whereas CHI3L1 deficiency markedly suppresses tumor formation. However, its involvement appears to be context-dependent and varies among different epithelial tumor types. These findings suggest that CHI3L1 is a potential therapeutic target and diagnostic biomarker for inflammation-associated cancers. Animal studies provide valuable insights into the immunological mechanisms of CHI3L1-mediated tumorigenesis but also highlight the need for cautious interpretation due to inherent technical limitations.

## 1. Introduction

The association between chronic inflammation and cancer development was first proposed by the German pathologist Rudolf Virchow in the mid-19th century [1]. He suggested that persistent inflammation resulting from tissue damage could create a microenvironment conducive to tumorigenesis. Inflammatory cells actively release factors that promote cell proliferation, and when inflammation becomes chronic, it can lead to malignant transformation at the affected site. Chronic inflammation in specific organs, such as the stomach, colon, and lungs, has been shown to significantly increase the risk of cancer in those tissues [2,3,4,5,6,7]. To sustain their growth, tumors manipulate the surrounding microenvironment by recruiting macrophages and fibroblasts to form supportive scaffolds. They also secrete inflammatory cytokines and growth factors that promote angiogenesis and nutrient supply [8,9].

Inflammation is now recognized as a hallmark of cancer, with a complex and multifaceted role in carcinogenesis. The initiation and progression of inflammation-associated cancers are closely linked to various factors, including chronic viral or bacterial infections, persistent tissue injury, autoimmune diseases, obesity, dysbiosis, environmental or chemical exposures, and genetic predisposition.

Chitin is a linear polysaccharide composed of β-1,4-linked N-acetylglucosamine units. It is a widely distributed structural polysaccharide found in the exoskeletons of insects and crustaceans, the cell walls of fungi, and the cuticles of nematodes [10,11]. Although vertebrates, including mammals, are unable to biosynthesize chitin, they do express chitinases and chitinase-like proteins (CLPs), which are thought to play important roles in immune responses, tissue remodeling, and inflammation [12,13]. Chitinases possess enzymatic activity capable of degrading chitin, whereas CLPs lack such activity due to mutations in their catalytic domains. Despite the absence of enzymatic function, CLPs are highly expressed in various inflammatory and neoplastic conditions, suggesting that they may exert distinct non-enzymatic roles in disease pathogenesis [12,13,14,15].

The two major enzymatically active chitinases in mammals are chitotriosidase (CHIT1) and acidic mammalian chitinase (AMCase), both of which are capable of degrading chitin [14,15]. In contrast, CLPs are catalytically inactive pseudoenzymes that possess a chitinase-like domain but lack enzymatic activity, although they structurally belong to the chitinase family [16]. A representative CLP is chitinase 3-like 1 (CHI3L1, also known as YKL-40 in humans and Brp-39 in mice), a multifunctional glycoprotein implicated in inflammation, tissue repair, fibrosis, and tumor development [17].

CHI3L1 is upregulated in response to pro-inflammatory cytokines and widely used as a clinical biomarker for disease activity and prognosis [18]. It is predominantly expressed by various cell types, including macrophages, neutrophils, epithelial cells, fibroblasts, and cancer cells, and it exerts its biological effects through interactions with the extracellular matrix and by activating multiple signaling pathways [18,19]. For example, CHI3L1 is induced during tissue injury and contributes to fibrosis by promoting fibroblast activation and collagen production [17]. It is strongly associated with fibrotic processes in chronic inflammatory diseases of the liver and lungs [20].

CHI3L1 modulates both innate and adaptive immune responses by influencing immune cell behavior in response to inflammatory stimuli [21]. Moreover, it activates key signaling pathways, such as STAT3 (signal transducer and activator of transcription 3) and AKT (protein kinase B), thereby enhancing cell survival and resistance to apoptosis [22]. Through these mechanisms, CHI3L1 promotes cancer cell proliferation, angiogenesis, invasion, and metastasis and contributes to the remodeling of the tumor microenvironment [23]. For example, in colorectal cancer, CHI3L1 expression has been shown to correlate with MAPK activation and poor clinical outcomes, and its overexpression significantly increases tumor growth and vascularization in xenograft models [23]. Additionally, by promoting the polarization of M2-type macrophages and establishing an immunosuppressive milieu, CHI3L1 supports sustained tumor growth [24].

Although CHI3L1 has been implicated in inflammation, fibrosis, and tumor progression in higher mammals, including humans, its regulatory mechanisms, functional roles, and interactions at the cellular and tissue levels remain incompletely understood [17,22]. To elucidate the role of CHI3L1 in the pathogenesis of inflammation-associated carcinogenesis, animal models that capture systemic and dynamic in vivo responses are indispensable. Murine models have proven to be particularly valuable tools for analyzing CHI3L1-mediated immunological effects, tissue remodeling, and tumor promotion. Phenotypic analyses using CHI3L1 knockout (KO) mice, overexpression models, and anti-CHI3L1 antibody interventions in both acute and chronic inflammatory conditions, such as allergic inflammation, colitis, and pulmonary fibrosis, have yielded critical insights [25,26,27].

This review aims to summarize and evaluate recent findings on the biological functions of CHI3L1 and its involvement in inflammation-associated carcinogenesis, based primarily on animal models. Special emphasis is placed on the role of CHI3L1 in epithelial dysplasia and cancer, with a focus on its potential as a diagnostic biomarker and therapeutic target.

## 2. Basic Biological Role of CHI3L1

CHI3L1 is a glycoprotein belonging to the glycoside hydrolase family 18 (GH18), and it has a molecular weight of approximately 40 kDa [13]. Structurally, it possesses a chitinase-like domain, but it is classified as a pseudo-chitinase as it lacks true enzymatic activity due to the absence of a critical catalytic residue (glutamic acid at position 140), which is required for chitin degradation [28]. Crystal structure analyses reveal that CHI3L1 can bind to polysaccharides, such as chitin, heparan sulfate, and collagen, as well as various extracellular matrix (ECM) components [29]. This binding is believed to mediate its physiological functions through interactions with signaling receptors, including IL-13 receptor alpha 2 (IL-13Rα2) and the receptor for advanced glycation end products (RAGE) [17,30].

Under physiological conditions, CHI3L1 is minimally expressed, but its expression is inducible under conditions associated with environmental stress, inflammation, tissue injury, fibrosis, and tumor progression [18]. High expression levels of CHI3L1 have been reported in various cell types and are regulated by specific cytokines, such as TNF, IL-6, IL-13, and TGF-β [17]. Conversely, CHI3L1 may also enhance the expression of pro-inflammatory cytokines, acting as a key mediator of chronic inflammation and fibrosis [31].

CHI3L1 is a multifunctional protein highly expressed in inflammatory diseases and within the tumor microenvironment, contributing to a range of physiological and pathological processes. It promotes cell proliferation, fibroblast activation, and collagen production, thereby facilitating wound healing and fibrotic remodeling [17]. It also regulates ECM dynamics through interactions with metalloproteases (MMPs) and TGF-β [32,33]. Notably, CHI3L1 is involved in both innate and adaptive immunity. For example, it has been reported to promote the polarization of M2-type macrophages, which support a Th2-dominant immune response and helps establish an immunosuppressive environment [22].

CHI3L1 activates intracellular signaling pathways, including MAPK (p42/p44 and p38), NF-κB, and PI3K/Akt, via binding to multiple cell surface receptors, such as IL-13Rα2, Galectin-3, and CD44v3. These pathways are involved in regulating cell survival, proliferation, migration, inflammation, fibrosis, and tumorigenesis [17,34,35,36]. CHI3L1 induces M2-type macrophages through the IL-13Rα2-mediated signaling cascade; in turn, these macrophages promote tumor cell proliferation, immunosuppression, and angiogenesis. This occurs through both autocrine and paracrine mechanisms, with CHI3L1 being secreted by both cancer cells and macrophages [35].

CHI3L1 specifically binds to IL-13Rα2 and activates the MAPK, PI3K/Akt, and Wnt/β-catenin pathways. These signals contribute to fibroblast activation, anti-apoptotic effects, and cell proliferation [21,22]. In addition, CHI3L1 binds to a receptor complex consisting of TMEM219 and CD44v3, activating pathways like AKT, MAPK (p42/p44), and β-catenin, thereby promoting tumorigenesis, particularly in cancer cells and cancer-associated fibroblasts [21,22,37,38]. Furthermore, interaction between CHI3L1 and Galectin-3, a lectin family protein, has been implicated in macrophages’ polarization and regulation, fibroblast activation, and the establishment of an immunosuppressive environment [20,35,39]. CHI3L1 has also been reported to promote inflammatory cytokine production through its interaction with RAGE, thereby contributing to chronic inflammation and the development of the tumor microenvironment [30,40]. CHI3L1 signaling via these receptors is often activated in distinct combinations depending on cell type and pathological context, and it serves as a key regulatory molecule in fibrosis, allergic inflammation, autoimmune diseases, and cancer [37,38,39,40,41].

## 3. Potential Role of CHI3L1 in Inflammation-Associated Cancer Development

Many pathogenic factors are involved in inflammation-associated cancer, as summarized in Table 1 [42,43,44,45,46,47,48,49,50,51,52,53,54,55,56,57,58,59,60,61,62,63,64]. CHI3L1 has received increasing attention as an in vivo molecule that links chronic inflammation to tumorigenesis. It is highly expressed in several major cancers and circulating tumor cells, suggesting its potential role as a prognostic biomarker in the clinical setting. Several studies have demonstrated that serum CHI3L1 levels and its expression in tumor tissues are significantly elevated in colorectal cancer compared to colitis without cancer [65]. Elevated serum CHI3L1 levels are positively associated with tumor progression, metastasis, recurrence rate, and reduced overall survival [19,65]. These findings underscore its relevance as a marker for disease severity and prognosis.

Notably, CHI3L1 plays a significant role in colitis-associated cancer, as suggested by findings from animal models of IBD [23]. It appears to serve as a molecular bridge between a sustained inflammatory milieu and tumor development, partly by shaping the tumor microenvironment through interactions with profibrotic factors and immunosuppressive cytokines [66]. This supports the notion that CHI3L1 functions not only as a biomarker but also as an active driver of inflammation-induced carcinogenesis.

In hepatocellular carcinoma (HCC), CHI3L1 is highly expressed in underlying liver pathologies, such as chronic hepatitis and liver fibrosis. Elevated serum CHI3L1 levels have been associated with an increased risk of HCC development and poor prognostic outcomes [67,68]. Similarly, in lung cancer, particularly non-small cell lung cancer, CHI3L1 expression is elevated in both serum and tumor tissue. This upregulation is closely linked with smoking history and chronic pulmonary inflammation [40,69]. High CHI3L1 expression levels in lung tissue have been correlated with treatment resistance, increased recurrence risk, and unfavorable clinical outcomes [40].

Taken together, although CHI3L1 is virtually absent in healthy individuals, it is markedly upregulated in various cancer types, where it plays a multifaced role. These roles include the recruitment of tumor-associated macrophages (TAMs), the promotion of immunosuppressive microenvironments, and the facilitation of tumor progression [64,70]. Despite these advances, the precise molecular mechanisms through which CHI3L1 drives tumorigenesis remain incompletely understood, warranting further investigation.

## 4. Animal Models of CHI3L1-Related Cancer

CHI3L1 is a molecule that plays a central role in the crosstalk between cancer and inflammation, and its functional significance has been clarified through studies using genetically engineered or chemically induced mouse models. These models provide critical insights into the pathophysiological roles of CHI3L1 in human diseases by capturing key aspects of inflammation-associated tumorigenesis and immune modulation.

### 4.1. CHI3L1 Overexpression Models

Several studies have reported that pulmonary melanoma metastasis is primarily mediated by IL-13Rα2-dependent mechanisms involving TGF-β1 production in the lung [22,71,72]. He et al. demonstrated that transgenic (Tg) mice constitutively overexpressing CHI3L1 exhibited enhanced tumor growth and immune evasion by promoting angiogenesis, activating cancer-associated fibroblasts (CAFs), and inducing M2-type macrophages [22]. In this model, Tg mice in which human CHI3L1 was selectively overexpressed in the lung showed significantly increased lung metastasis of injected melanoma cells compared to wild-type (WT) controls [22]. Notably, the study emphasized the critical role of the IL-13-CHI3L1-IL-13Rα2 axis in malignant melanoma progression. This conclusion was supported by the finding that CHI3L1-driven metastasis was dramatically reduced in IL-13Rα2 KO mice, confirming that IL-13Rα2 is essential for CHI3L1-mediated tumor progression in this model [22].

In contrast, Zhang et al. reported that CHI3L1 overexpression suppressed melanoma growth in a subcutaneously injected melanoma mouse model [73]. Melanoma cells are highly immunogenic, and tumor development and metastasis are often associated with impaired immune responses. Although elevated serum CHI3L1 levels have been linked to increased mortality in melanoma patients, the primary sources of CHI3L1 are immune cells, such as macrophages and neutrophils, rather than tumor cells themselves [73,74,75]. Interestingly, CHI3L1 overexpression in the murine model was associated with increased expression and activation of anti-tumor T-cell-related genes, suggesting that CHI3L1-targeted therapy might synergize with immune checkpoint inhibitors [73] (Figure 1). In fact, CHI3L1 may paradoxically promote tumor progression by stimulating the PD-1/PD-L1 axis. Therefore, bispecific antibodies targeting both PD-1/PD-L1 and CHI3L1 represent a promising therapeutic strategy for pulmonary metastasis and melanoma progression [37] (Figure 1).

MOLF/EiJ is a unique wild-derived strain that naturally expresses high levels of CHI3L1 in both serum and colonic epithelial cells (CECs) due to nucleotide polymorphisms in the proximal promoter regions of the CHI3L1 gene (−1 to −517 bp) [76]. Under steady-state conditions, MOLF/EiJ mice exhibit colonic epithelial hyperproliferation and polyp-like nodules without cytological abnormalities or neoplastic changes. However, after repeated weekly injections of the carcinogen Azoxymethane (AOM), CHI3L1-expressing CECs in MOLF mice show accelerated tumor growth and dysplastic changes compared to those in B6 WT mice. These findings suggest that CHI3L-overexpressing CECs are highly proliferative and prone to malignant transformation [76].

CHI3L1 is also known to induce PD-L1 expression in lung macrophages, thereby promoting immune evasion. In CHI3L1-transgenic mice, the number of CD45^+^ CD11b^+^ CD68^+^ PD-L1^+^ cells increased, and this effect was suppressed by the administration of anti-CHI3L1 antibodies [37,77]. Moreover, CHI3L1 modulates the expression of T cell costimulatory molecules (ICOS, CD28) and immunosuppressive molecules (CTLA-4, B7-1, B7-2), acting as a broad regulator of immune responses [77]. A study by Zhou et al. also demonstrated that CHI3L1 overexpression promoted an increase in M2-type macrophages (Arg^+^, CD206^+^) within the tumor microenvironment [35].

### 4.2. CHI3L1 Knockout Models

In the AOM and dextran sulfate sodium (DSS)-induced chronic-colitis-associated cancer model, Low et al. demonstrated the role of epithelial CHI3L1 in carcinogenesis using CHI3L1 KO mice. These mice exhibited reduced tumor incidence both in size and number compared to age-matched WT controls, with key findings summarized in Figure 2 [30]. Because CHI3L1 is involved in tissue restitution, CHI3L1 KO mice showed significantly delayed repair of the colonic epithelium following DSS treatment. Furthermore, bone marrow chimera experiments revealed that carcinogenesis occurs after epithelial regeneration during recovery from chronic inflammation. CHI3L1 expression in non-hematopoietic cells, including epithelial and stromal cells, was essential for the neoplastic transformation of the colonic epithelium [30].

CHI3L1-mediated proliferation and survival of CECs appear to suppress S100A9, a pro-apoptotic protein that is highly expressed during the acute phase of inflammation. Both CHI3L1 and S100A9 compete for binding to RAGE. In the chronic phase of colitis, a CHI3L1^high^ S100A9^low^ environment promotes tumor cell survival in the inflamed mucosa [30].

CHI3L1 is also known to promote lung cancer development by inhibiting the tumor suppressor protein p53. In CHI3L1 KO mice, lung tumor metastasis was significantly reduced, along with increased expression of p53, p21, BAX, and cleaved caspase-3 [78]. Conversely, the phosphorylation of STAT3 and the expression of cell-cycle-associated proteins, such as Cyclin E1 and CDK2, were decreased, suggesting that CHI3L1 suppresses the p53 pathway to promote tumor progression [78].

In liver cancer models using carbon tetrachloride (CCl_4_) and diethyl-nitrosamine (DEN), CHI3L1 facilitates the progression from chronic injury to fibrosis and hepatocellular carcinoma (HCC) [22]. It activates hepatic stellate cells, induces collagen and TGF-β production, and enhances fibrogenesis through IL-13Rα2 and Galectin-3 signaling [22].

Furthermore, CHI3L1 contributes to an immunosuppressive tumor microenvironment by promoting M2-type macrophage polarization, suppressing T cell activity, and increasing levels of IL-10 and VEGF (vascular endothelial growth factor). This suppresses immune clearance of abnormal cells and accelerates carcinogenesis. In CHI3L1 KO mice, liver fibrosis and tumor number and size were significantly reduced, while T cell infiltration and immune activation were enhanced, indicating dual roles of CHI3L1 in fibrosis and carcinogenesis [22].

### 4.3. Xenograft and Syngeneic Tumor Models

CHI3L1 has been reported to stimulate the proliferation of fibroblasts, synovial cells, and chondrocytes by modulating chemokine and MMP expression under inflammatory conditions, suggesting its role in the recruitment of stromal cells into the tumor microenvironment [79,80,81].

In 2012, Kawada et al. demonstrated that CHI3L1 overexpression in colon cancer cells significantly enhanced tumor growth in a murine xenograft model by promoting proliferation, tumor-associated macrophages’ (TAMs) recruitment, and angiogenesis [23]. These findings support that CHI3L1 fosters tumor development by creating a supportive microenvironment. It enhances IL-8 and MCP-1 secretion via ERK and JNK signaling activation [23]. Findings in synergistic mouse models confirm that CHI3L1 promotes tumor progression and immunosuppression even in immunocompetent settings, parallelling xenograft observations [82].

Chen et al. intravenously injected human breast cancer cell lines (MDA-MB-231 and MDA-MB-435) into nude mice and treated them with recombinant CHI3L1 or PBS. CHI3L1 treatment significantly increased lung metastasis compared to PBS controls. CHI3L1 binds to IL-13Rα2 on cancer cells and activates MAPK (ERK and JNK) signaling, upregulating c-Fos, c-Jun, and MMP-9, which promotes breast cancer metastasis [83].

Another study implanted breast cancer stem cells (BCSCs-231) into the mammary fat pad of mice and administered anti-CHI3L1 antibody, which suppressed tumor growth. This effect was linked to CHI3L1-driven activation of the MAF/CTLA-4 pathway, reducing CD8^+^ T cell infiltration and promoting immune evasion [84].

In a model using 4T1 breast cancer cells, CHI3L1 expression was upregulated during the premetastatic phase. It promoted angiogenic factors (CCL2, CXCL2, and MMP-9) in newly migrating tumor cells. CHI3L1 acted on CD11b^+^ Gr1^+^ macrophages to establish a pro-metastatic lung niche [85].

Additionally, intraperitoneal injection of chitin microparticles (<10 μm) reduced CHI3L1 and pro-angiogenic factors in the lung, suggesting that CHI3L1 inhibition may block metastasis [85,86]. These findings highlight CHI3L1’s role in breast cancer progression via immunoregulation, angiogenesis, and matrix remodeling, supporting its candidacy as a potential therapeutic target.

### 4.4. CHI3L1-Driven Inflammation-Associated Cancer Models

As noted in the previous section, a study using the AOM/DSS model revealed that CHI3L1 KO mice developed more severe colitis than WT mice but exhibited a significantly lower tumor incidence. This suggests that CHI3L1 promotes intestinal epithelial cell proliferation during the chronic inflammatory phase, thereby contributing to tumorigenesis. Additionally, CHI3L1 appears to facilitate inflammation-associated carcinogenesis by downregulating S100A9 expression and promoting epithelial cell survival and proliferation [30]. Bone marrow transplantation experiments demonstrated that the tumorigenic effects of CHI3L1 primarily depend on its expression in non-hematopoietic cells, particularly intestinal epithelial cells [30].

Another study reported that CHI3L1 mRNA expression increased with tumor progression in the AOM/DSS model, with particularly high levels in the distal colon [87]. Furthermore, in vitro experiments using the HT29 human colon cancer cell line showed that CHI3L1 elevated intracellular reactive oxygen species (ROS) in the presence of hydrogen peroxide. These findings suggest that CHI3L1 may promote tumor development by inducing DNA damage through oxidative stress [87]. Taken together, studies using the AOM/DSS-induced colon cancer model demonstrate that CHI3L1contributes to colon carcinogenesis by enhancing epithelial proliferation, increasing oxidative stress, and establishing an immunosuppressive microenvironment [30,87]. These findings highlight CHI3L1 as a promising therapeutic target in colon cancer, warranting further investigation.

In an inflammation-induced lung cancer model, CHI3L1 expression was significantly upregulated and positively correlated with tumorigenesis [88]. In this context, CHI3L1 acts as a downstream effector of STAT3 and promotes tumor progression by enhancing the production of inflammatory cytokines.

In a mouse model of Lewis lung carcinoma, administration of an anti-CHI3L1 antibody significantly suppressed tumor growth and metastasis. This effect was associated with STAT6-dependent M2-type macrophages’ polarization, indicating that CHI3L1 contributes to immunoregulation and immunosuppression within the tumor microenvironment [89].

In summary, CHI3L1 plays a central role in inflammation-associated cancers. It is induced by inflammatory cytokines and growth factors and secreted by both tumor cells and surrounding immune cells. CHI3L1 interacts with multiple receptors, including IL-13Rα2, RAGE, CD44, and Galectin-3, and it activates downstream pathways, such as PI3K/Akt, NF-κB, and TGF-β/Smad. Through these mechanisms, CHI3L1 exerts diverse tumor-promoting effects, including enhanced cell proliferation, immunosuppression, angiogenesis, and metastasis. Its pathophysiological relevance has also been demonstrated across various cancer types, particularly colorectal and lung cancer. The therapeutic potential of CHI3L1-targeted strategies, including anti-CHI3L1 antibodies, has also been reported [37,57,77]. As a key mediator of the inflammation–cancer axis, CHI3L1 is emerging as a novel and promising target for cancer therapy.

## 5. Translational Implications as Clinical Biomarkers and Potential Therapeutic Target

CHI3L1 is gaining increasing attention as a potential biomarker and therapeutic target in inflammation-associated cancers, as summarized in Table 2 [90,91,92,93,94,95,96,97,98,99,100,101,102,103,104,105,106,107,108,109,110,111,112,113,114,115,116,117]. In a mouse model of AOM/DSS-induced colitis-associated cancer, CHI3L1 expression is significantly upregulated in intestinal epithelial cells, contributing to tumor initiation and progression [30]. In both this animal model and human patients with IBD-associated dysplasia or cancer, fecal CHI3L1 levels increase in parallel with inflammatory progression and positively correlate with the onset of colitis-associated cancer. These findings suggest its utility as a non-invasive biomarker for detecting dysplasia or early-stage tumorigenesis in IBD [30].

In contrast, other studies have reported that fecal CHI3L1 is not a reliable marker of colorectal cancer detection in symptomatic patients in primary care settings [90]. Specifically, no significant differences were observed in fecal CHI3L1 concentrations between patients and healthy controls (*p* = 0.193), indicating low diagnostic performance. Moreover, combining CHI3L1 with fecal occult blood testing did not enhance diagnostic accuracy [90]. These discrepancies may arise from heterogeneity in the studied populations, including sporadic populations, as well as differences in lesion progression and inflammatory status. Additional confounding factors include methodological variability in CHI3L1 quantification, cutoff values, and small sample sizes. Further studies are needed to clarify the diagnostic value of CHI3L1 by considering disease type and stage, inflammation status, and characteristics of the target population. A better understanding of CHI3L1 kinetics and function in distinct clinical contexts, such as chronic inflammation versus sporadic colorectal cancer, may help define its utility as a context-specific biomarker.

Chen et al. demonstrated that CHI3L1 expression is elevated even in the non-dysplastic mucosa in UC patients with dysplasia, suggesting its potential as a predictor of future tumorigenesis [91]. Additionally, serum CHI3L1 levels have been shown to correlate with poor prognosis, metastatic potential, and treatment response in colorectal cancer patients [92]. These findings highlight the critical involvement of CHI3L1 in colitis-associated cancer development and progression and emphasize the relevance of its expression dynamics for early diagnosis and therapeutic design.

High CHI3L1 expression has also been reported in lung cancer tissues and patient serum, where levels increase with disease stage and correlate with poor prognosis [93]. Significantly elevated serum CHI3L1 has been observed across major lung cancer subtypes, including adenocarcinoma, squamous cell carcinoma, and small cell lung carcinoma [93]. Meta-analyses further support these finding, showing that high CHI3L1 expression is significantly associated with reduced overall survival in lung cancer patients.

In inflammation-associated cancers (Table 2), CHI3L1 expression in tumor tissues is consistently associated with increased tumor aggressiveness, higher histological grade, advanced clinical stage, and poorer prognosis [94,97,101,104,105,106,109,114,115]. Similarly, elevated serum CHI3L1 levels are reported to serve as independent prognostic indicators, correlating with shorter overall survival, reduced recurrence-free survival, and more advanced disease states [36,92,93,95,96,102,103,107,108,110,114,116]. However, the diagnostic specificity of CHI3L1 remains limited in several tumor types [90,95,100]. Interestingly, urinary CHI3L1 levels may help distinguish between high- and low-grade tumors in bladder cancer, suggesting its potential as a non-invasive diagnostic biomarker in this context [115].

In Pten^flox/flox^ mice, non-steroidal anti-inflammatory drugs (NSAIDs), such as aspirin and naproxen, inhibited the growth of TMPRSS2-ERG fusion gene-driven prostate tumors. Among seven tumor-promoting inflammatory molecules significantly reduced in plasma and prostate tissue, CHI3L1 was identified as a key mediator [113]. These results suggest that CHI3L1 contributes to tumor proliferation and inflammation in immune cell-rich prostate cancer, further supporting its relevance as a biomarker.

In hepatocellular carcinoma (HCC), serum CHI3L1 does not appear to be a reliable diagnostic marker, as it cannot effectively distinguish HCC from liver cirrhosis [95]. Similarly, in pancreatic cancer (PC), plasma CHI3L1 alone lacks sufficient diagnostic power [100]. However, when combined with established biomarkers, such as serum CA 19-9 and plasma IL-6, CHI3L1 may help identify early-stage PC patients with poor prognosis [100]. These results suggest that CHI3L1, when used in combination with other markers, may improve diagnostic accuracy and facilitate subtypes stratification in selected cancers.

Furthermore, elevated serum CHI3L1 has been shown to serve as an independent prognostic factor for both overall survival and recurrence-free survival in HCC patients undergoing curative resection [95]. In metastatic PC, plasma CHI3L1 may also act as a prognosis biomarker in patients receiving immune checkpoint inhibitors and radiotherapy [98].

Taken together, these findings indicate that CHI3L1 plays a central role in the progression and of multiple inflammation-associated cancers and may function both as a biomarker and a therapeutic target. From a therapeutic standpoint, CHI3L1 inhibition may suppress tumor growth, and therapeutic strategies, including anti-CHI3L1 monoclonal antibodies and bispecific antibodies targeting CHI3L1/PD-1, are currently under development (Figure 1) [37,57].

Nevertheless, translating animal data to human applications presents several challenges. Differences in immune system architecture and tumor microenvironments between mice and humans, as well as disparities in the nature of inflammation and cancer progression, may limit the applicability of animal model findings. In addition, commonly used models rely on repeated acute inflammation, which may not fully replicate the complex chronic pathophysiology of human disease. Therefore, when assessing CHI3L1’s role of the efficacy of therapeutic interventions, it is essential to account for these limitations and compliment animal studies using huma cells, tissues (e.g., cancer cell lines), and clinical specimens.

## 6. Future Perspectives and Conclusions

Studies using animal models of inflammation-associated cancer have shown that CHI3L1 is not merely a marker of inflammation but also an active driver of tumor development [17,22,25,27,30,91]. In particular, findings from murine models have demonstrated that CHI3L1 contributes to several key processes involved in the establishment and maintenance of the tumor microenvironment, including angiogenesis, tumor cell proliferation, immunosuppression, and polarization of macrophages toward an M2 phenotype. Building on these experimental insights, future research should aim to elucidate the intracellular and extracellular signaling pathways mediated by CHI3L1 in greater detail. Moreover, given that current mouse models do not fully recapitulate human disease pathology, the development of more clinically relevant models will be essential.

To evaluate the therapeutic potential of the CHI3L1-targeted strategy, including humanized anti-CHI3L1 antibodies, comprehensive preclinical and clinical studies are urgently needed. Several preclinical investigations have already demonstrated the anti-tumor efficacy of CHI3L1-neutralizing antibodies in models of lung cancer and melanoma [37,77]. While no clinical trials have been published or registered to date, these findings highlight the translational promise of CHI3L1-targeted therapies. CHI3L1 holds potential not only as a diagnostic and prognostic biomarker but also as a therapeutic target that bridges chronic inflammation and cancer. It is expected to emerge as a key molecule in both future cancer research and clinical applications. However, challenges remain. These include the risk of off-target effects, inter-individual variability in biomarker expression levels, and the need to more precisely define cancer types, such as colorectal cancer and lung cancer, in which CHI3L1-targeted approaches are most likely to provide therapeutic benefit.

## Figures and Tables

**Figure 1 cells-14-00982-f001:**
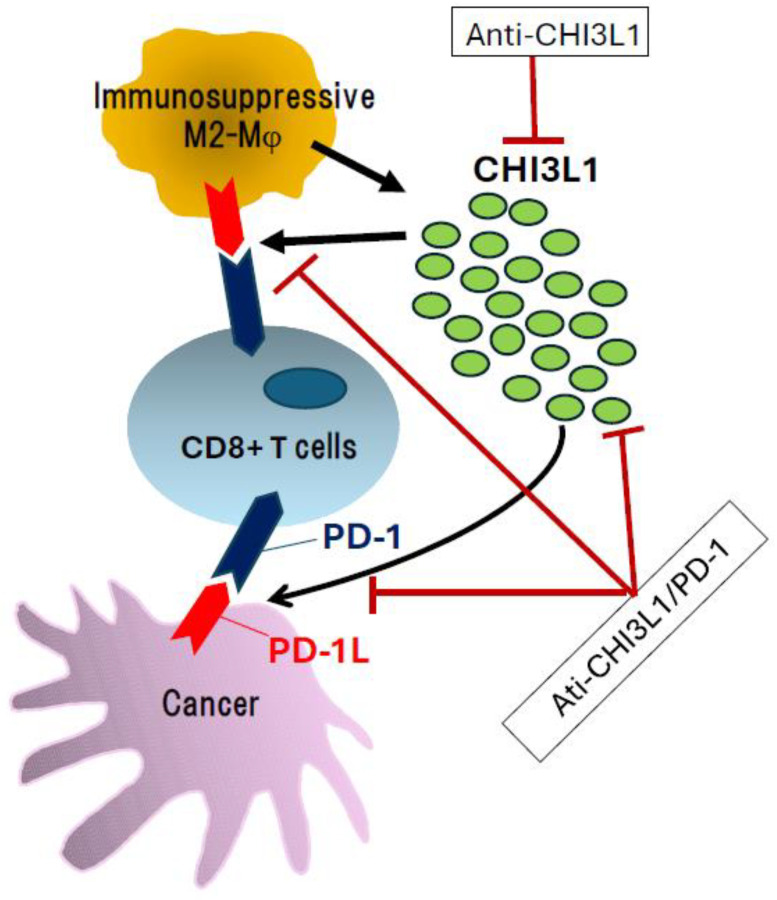
Influence of CHI3L1 and its inhibition with specific antibodies. Overexpression and secretion of CHI3L1 are associated with enhanced anti-tumor immune responses through stimulation of the PD-1/PD-L1 axis. In this context, targeting CHI3L1 using anti-CHI3L1 antibodies or bispecific antibodies against both CHI3L1 and PD-1 represents a promising therapeutic strategy for cancer treatment.

**Figure 2 cells-14-00982-f002:**
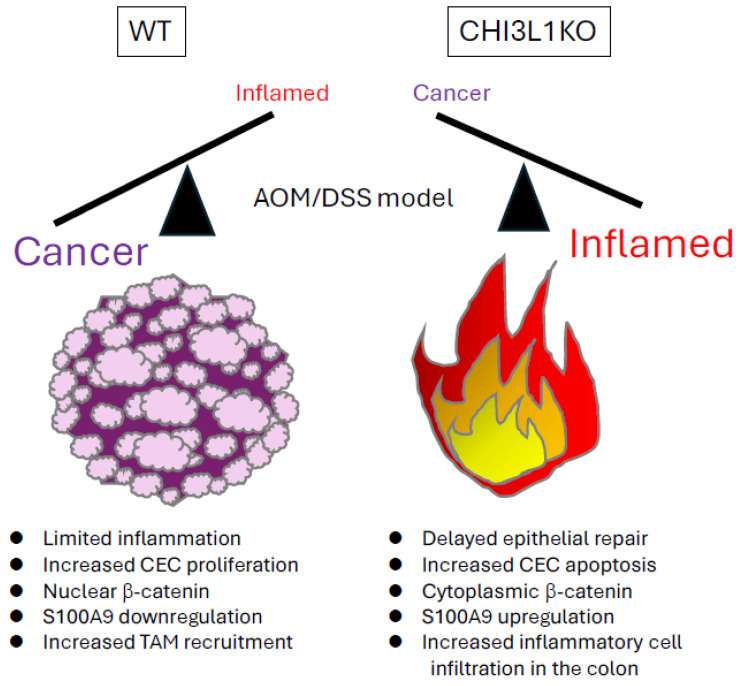
AOM/DSS-induced colitis in wild-type (WT) and CHI3L1 knockout (KO) mice. In this chronic-colitis-associated cancer model, CHI3L1 KO mice showed reduced tumor burden, with smaller and fewer tumors, compared to WT controls. However, CHI3L1 KO mice also exhibited delayed epithelial repair, prolonged colonic inflammation, increased epithelial apoptosis, and greater inflammatory cells compared to WT mice.

**Table 1 cells-14-00982-t001:** Major pathogenic causes of inflammation-associated carcinogenesis.

Major Causes	Possible Mechanisms	Examples	Ref.
Infections	Persistent local infection-based chronic inflammation, oxidative stress, and DNA damage induce cancer formation.	HCV, HBV → HCC EBV → Head and neck cancer HPV → Cervical cancer H. pylori → Gastric cancer	[42,43,44]
Tissue injuries	Recurrent tissue damage causes compensatory tissue growth and inflammation.	IBD → Colorectal cancer Barrett esophagus → Esophageal cancer LC → HCC Chronic pancreatitis → Pancreatic cancer	[45,46,47,48]
Autoimmune-related disorders	Autoimmune-associated inflammation causes tissue damage and tumor formation.	IBD → colorectal cancer Sjogren’s syndrome → Malignant lymphoma	[49,50,51]
Obesity	Excessive secretion of pro-inflammatory cytokines due to fatty tissue growth caused by obesity, creating a tumor-promoting environment.	Obesity → Breast cancer, colorectal cancer, pancreatic cancer	[52,53]
Dysbiosis	Intestinal dysbiosis overstimulates immune responses and promotes tumor formation.	Dysbiosis → HCC, colorectal cancer	[54,55]
Environmental factors	Additive exposure to harmful substances causes chronic inflammation and mutagenesis.	Asbestosis → Mesothelioma Silica → Lung cancer Tobacco smoke → Lung cancer, cancer of airways	[56,57]
Genetic factors	Promoting tumorigenesis due to abnormalities in the inflammatory signaling pathway(s).	IL-10 genetic mutation → EO- IBD	[58]
Aging	Aging-related decline in the immune system leads to chronic low levels of inflammation, creating a basis for cancer.	Aging → Prostate cancer, colon cancer	[59]
Micronutrient deficiency	Vitamin D or iron deficiency enhances inflammation.	Digestive cancer	[60,61]
Iatrogenic factors	Chemotherapy/radiation therapy induced inflammation	Combination therapy → Secondary lung cancer	[62]

Abbreviations: EO, early onset; EBV, Epstein–Bar virus; H. pylori, Helicobacter pylori; HBV, hepatitis B virus; HCC, hepatocellular carcinoma; HCV, hepatitis C virus; HPV, human papilloma virus; IBD, inflammatory bowel disease; LC, liver cirrhosis; SCC, squamous cell carcinoma.

**Table 2 cells-14-00982-t002:** CHI3L1 as a biomarker in inflammation-associated cancer.

Cancer	Samples (Relative, Increase, X-Fold)	Results	Ref.
Colon	Stool (<1.3-fold)	・ Fecal CHI3L1 levels increase with progression of inflammation in CAC. ・Fecal CHI3L1 is not a reliable biomarker of colonic lesions in symptomatic primary care patient.	[50,90]
	Tissue (<25-fold)	・ Significantly increased CHI3L1 expression in non-dysplastic mucosa from patients with IBD who had dysplasia/adenocarcinoma compared with control individuals.	[91]
	Serum (<2-fold)	・ Patients with high operative serum CHI3L1 concentration had significantly shorter survival than patients with normal CHI3L1.	[92]
Lung	Tissue (<4-fold) Serum (<1.5-fold)	・ CHI3L1 is highly expressed in lung cancer tissue and in the serum of patients with poor prognosis as well as animal models.	[88,93]
Liver	Tissue (<16-fold)	・ Elevated tissue CHI3L1 in HCC compared to adjacent peritumoral tissues and further elevated in tumors with metastasis. ・Positive CHI3L1 expression was significantly associated with clinicopathological features in HCC. ・HCC patients with positive CHI3L1 expression were correlated with poor overall survival and disease-free survival.	[94]
	Serum (<3-fold)	・ Serum CHI3L1 may not be as reliable a biomarker for HCC. ・Serum CHI3L1 may act as an independent prognostic factor for overall and RFS in HCC patients receiving curative resection.	[95,96]
Stomach	Tissue (<2.5-fold)	・ CHI3L1 expression was significantly higher in gastric cancer tissues compared to adjacent nonneoplastic tissues. ・Elevated CHI3L1 was positively correlated with the poor prognosis and aggressive behavior of gastric cancer cells.	[97]
	Serum (<4-fold)	・ Elevated serum CHI3L1 was associated with invasion depth, lymph node status, and tumor stage in patients with gastric cancer.	[36]
Pancreas	Plasma (<3-fold)	・ Increased plasma CHI3L1 is related to shorter overall survival (OS) in mPC (metastatic pancreatic cancer) patients. ・Plasma CHI3L1 could be a biomarker in patients with mPC receiving ICIs (immune checkpoint inhibitors) with radiotherapy. ・Plasma CHI3L1 in combination with serum CA 19-9 and plasma IL-6 could be useful to identify a subgroup of low-stage PC patients.	[98,99,100]
Breast	Tissue	・ Elevated expression levels of CHI3L1 correlate with tumor grade and poor differentiation.	[101]
	Serum (<2-fold)	・ Increased serum CHI3L1 levels in metastatic breast cancer patients. ・Elevated CHI3L1 levels in patients with locally advanced breast cancer.	[102,103]
Cervix	Tissue (<3.4-fold)	・ High levels of CHI3L1 in CSCC (cervical squamous cell carcinoma). ・Elevated CHI3L1 in invasive CxCa (cervical cancer). ・Elevated CHI3L1 mediates VM (vasculogenic mimicry) in CxCa.	[104,105,106,107,108]
	Serum (<5.1-fold)	・ Elevated serum CHI3L1 in both SCC (squamous cell carcinoma) and adenocarcinoma. ・Elevated serum CHI3L1 was associated with shorter RFS (recurrence-free survival) and OS (overall survival). ・Early changes in serum CHI3L1 levels may serve as a biomarker to monitor patients with CxCa after the operation and other therapies.	
Ovary	Tissue (<38-fold)	・ Tissue CHI3L1 was closely correlated with the clinical stage of EOC (epithelial ovarian cancer). ・Elevated tissue CHI3L1 correlated with significantly shorter overall survival time in OC patients.	[109]
	Serum (<2-fold)	・ Elevated serum CHI3L1 in OC patients regardless of tumor grade, histology, or patient age. ・The serum levels of CHI3L1 in early-stage patients may predict disease recurrence and survival. ・Elevated serum CHI3L1 in early-stage OC patients.	[110]
	Plasma (<18-fold)	・ Plasma CHI3L1 levels are associated with the stage and prognosis in OC. ・Elevated plasma CHI3L1 at the time of relapse of OC.	[111,112]
Prostate	Tissue (<5-fold)	・ CHI3L1 is one of the seven significantly suppressed tumor promoting inflammatory molecules identified in the plasma and prostate sample after an NSAID-treated prostate cancer model.	[113]
Bladder	Tissue (<3.4-fold)	・ Elevated tissue CHI3L1 was significantly associated with aggressive clinicopathological features in UTUC or UBUC. ・Elevated tissue CHI3L1 can serve as an independent prognostic factor for worse DSS and MFS in both UTUC and UBUC groups.	[114,115,116]
	Serum (<2.2-fold)	・ Elevated serum CHI3L1 in patients with BC (Bladder Cancer) associated with poor survival. ・Elevated serum CHI3L1 as an independent prognostic factors in BC.	
	Urine (<23.2-fold)	・ Urine CHI3L1 levels can differentiate invasiveness in BC patients.	
Kidney	Blood (<2.4-fold)	・ High blood CHI3L1 levels are associated with poor survival in patients with renal cell carcinoma.	[117]

Abbreviations: BC, bladder cancer; CAC, colitis-associated cancer; CA 19-9, carbohydrate antigen 19-9; CSCC, cervical squamous cell carcinoma; CxCa, cervical cancer; EOC, epithelial ovarian cancer; HCC, hepatocellular carcinoma; IBD, inflammatory bowel disease; ICIs, immune checkpoint inhibitors; IL-6, interleukin-6; mPC, metastatic pancreatic cancer; MVD, microvessel density; MFS, Metastasis-Free Survival; NSAID, non-steroidal anti-inflammatory drugs; OS, overall survival; OC, ovarian cancer; RFS, recurrence-free survival; SCC, squamous cell carcinoma; TNM, tumor, nodes, metastasis; UTUC, upper tract urothelial carcinoma; UBUC, urinary bladder urothelial carcinoma; VEGF, vascular endothelial growth factor; VM, vasculogenic mimicry.

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
