# Peer review of "Future Perspectives and Conclusions from Animal Models of CHI3L1-Related Inflammation-Associated Cancer"

_cells, 2025, doi:10.3390/cells14130982_

Round 1
Reviewer 1 Report
Comments and Suggestions for Authors
This manuscript offers a timely and comprehensive view of the role of CHI3L1 in linking chronic inflammation and cancer. The authors have successfully summarized the current knowledge from animal models and highlight the significance of CHI3L1 as both a biomarker and a potential therapeutic target in oncology. The paper is well-structured, clear and insightful.
However, following revisions need to be made by the authors:
- In the abstract state whether this is a universal phenomenon across all epithelial tumors or if there are context-specific mechanisms.
- In abstract, add a brief mention of the signaling mechanisms or cellular components involved in CHI3L1’s effects to help understand how CHI3L1 contributes to tumor progression.
- Lines 42-47: Add a comparison with chitinase-active vs chitinase-like proteins to help understand their respective roles in disease context.
- Lines 64-71: The signaling mechanisms (STAT3, AKT, MAPK) and effects (angiogenesis, migration, resistance) are well described. Add an example or a key study demonstrating their role in a particular tumor context (such as lung or colon).
- Lines175-178: Add a brief sentence stating how these mouse models collectively aid in understanding CHI3L1’s role in human disease.
- Lines180-188: Add a comparison to a similar overexpressing mouse model (with or without IL-13Rα2) to clarify whether IL-13Rα2 is essential for melanoma progression.
- Lines274-280: Add a comparison to a syngeneic mouse model to highlight whether human xenografts reflect mechanisms relevant to human disease progression.
- Line 351-352: Provide a comparison to normal controls or a quantitation ( such as fold-upregulation) if available.
- Line 441: Rename title as “Future perspectives and conclusions”
- Line 444: Specify examples of murine models.
- Line 452-453: Provide a reference to any ongoing trials or preclinical data on these antibodies if available.
- Line 454-455: Briefly address challenges such as off-target effects, biomarker variability or specify cancer types with greatest promise.

Author Response
Reviewer 1:
This manuscript offers a timely and comprehensive view of the role of CHI3L1 in linking chronic inflammation and cancer. The authors have successfully summarized the current knowledge from animal models and highlighted the significance of CHI3L1 as both a biomarker and a potential therapeutic target in oncology. The paper is well-structured, clear and insightful.
However, the following revisions need to be made by the authors:
Comment-1) In the abstract state whether this is a universal phenomenon across all epithelial tumors or if there are context-specific mechanisms.
Response-1)Thank you for the comment. In abstract, we added the following sentence ”However, the involvement of CHI3L1 is context-dependent and varies among different types of epithelial tumors“
Comment-2) In abstract, add a brief mention of the signaling mechanisms or cellular components involved in CHI3L1’s effects to help understand how CHI3L1 contributes to tumor progression.
Response-2) We have added the following sentence, “Mechanistically, CHI3L1 exerts its effects by modulation of STAT3, MAPK, and PI3K/Akt signaling pathways, and by interacting with cell surface receptors such as IL-13Ra2 and RAGE”.
Comment-3) Lines 42-47: Add a comparison with chitinase-active vs chitinase-like proteins to help understand their respective roles in disease context.
Response-3) Thanks for the helpful comment. We have added the following sentences as a comparison with chitinase-active vs chitinase-like proteins. “Chitinases possess enzymatic activity capable of degrading chitin, whereas CLPs lack such activity due to mutations in their catalytic domains. Despite the absence of enzymatic function, CLPs are highly expressed in various inflammatory and neoplastic conditions, suggesting that they may exert distinct non-enzymatic roles in disease pathogenesis [12-15]”.
Comment-4) Lines 64-71: The signaling mechanisms (STAT3, AKT, MAPK) and effects (angiogenesis, migration, resistance) are well described. Add an example or a key study demonstrating their role in a particular tumor context (such as lung or colon).
Response-4) We have added an example as follows: “For example, in colorectal cancer, CHI3L1 expression has been shown to correlate with MAPK activation and poor clinical outcomes, and its overexpression significantly increases tumor growth and vascularization in xenograft models [23]”.
Comment-5) Lines175-178: Add a brief sentence stating how these mouse models collectively aid in understanding CHI3L1’s role in human disease.
Response-5) Thanks for your valuable suggestion. We have added the following sentence: “These models provide critical insights into the pathophysiological roles of CHI3L1 in human diseases by capturing key aspects of inflammation-associated tumorigenesis and immune modulation”.
Comment-6) Lines180-188: Add a comparison to a similar overexpressing mouse model (with or without IL-13Rα2) to clarify whether IL-13Rα2 is essential for melanoma progression.
Response-6) We have added a comparison as follows: “This conclusion was supported by the finding that CHI3L1-driven metastasis was dramatically reduced in IL-13Ra2 KO mice, confirming that IL-13Ra2 is essential for CHI3L1-mediated tumor progression in this model [22]”.
Comment-7) Lines274-280: Add a comparison to a syngeneic mouse model to highlight whether human xenografts reflect mechanisms relevant to human disease progression.
Response-7)We have added a comparison as follows: “Findings in synergistic mouse models confirm that CHI3L1 promotes tumor progression and immunosuppression even in immunocompetent settings, parallelling xenograft observations [82]”.
We have added the following new Reference #82
Yang, P.S.;Yu, M.H.; Hou, Y.; Chang, C.; Lin, S.C.; Kuo, I.Y.; Su, P.C.; Cheng, H.C.; Su, W.C.; Shan, Y.S.; et al. Targeting protumor factor chitinase-3-like-1 secreted by Rab37 vesicles for cancer immunotherapy. Theranostics 2022, 340, 340-361.:
Comment-8) Line 351-352: Provide a comparison to normal controls or a quantitation ( such as fold-upregulation) if available.
Response-8) Thank you for the valuable comment. According to the suggestion, we have added a comparison to normal controls or a quantitation in Table 2.
Comment-9) Line 441: Rename title as “Future perspectives and conclusions”
Response-9) Thank you for the suggestion. Now we have changed the title of this review article as follows: “Future perspectives and conclusions from animal models of CHI3L1-related inflammation-associated cancer”.
Comment-10) Line 444: Specify examples of murine models.
Response-10) We have added some selected references after the sentence.
Comment-11) Line 452-453: Provide a reference to any ongoing trials or preclinical data on these antibodies if available.
Response-11) Thank you for the insightful comment. We have added references to recent preclinical studies demonstrating the therapeutic efficacy of humanized anti-CHI3L1 antibodies in murine cancer models. While there are currently no ongoing clinical trials published or registered to our knowledge, these findings underscore the translational potential of this approach. We have revised the sentence accordingly on lines 455-459.
Comment-12) Line 454-455: Briefly address challenges such as off-target effects, biomarker variability or specify cancer types with greatest promise.
Response-12) Thank you for the valuable comment. We have now revised the text to acknowledge important challenges in the therapeutic target of CHI3L1 as follows: “However, challenges remain. These include the risk of off-target effects, inter-individual variability in biomarker expression levels, and the need to more precisely define the cancer types, such as colorectal cancer and lung cancer, in which CHI3L1-targeted approaches are most likely to provide therapeutic benefit”.
Reviewer 2 Report
Comments and Suggestions for Authors
The submitted manuscript addresses a relevant topic and presents findings that could contribute meaningfully to the field, congratulations to the authors! The manuscript demonstrates potential. However, in its current form, the manuscript requires significant technical and linguistic revision before it can be considered for publication.
Major Concerns:
-
Language and Grammar:
The text contains numerous grammatical and syntactical errors that hinder readability and overall clarity. A thorough language edit by a native or fluent English speaker is strongly recommended. -
Formatting Issues:
The font is not uniform throughout the document, which disrupts the visual consistency.
-
The title includes a full stop, which is not standard practice and should be removed.
-
Several typographical errors are present (e.g., inconsistent spacing, punctuation misuse, etc.).
-
-
Inaccurate Symbols and Units: In Table 2, the range should be replaced with a proper hyphen: “98–100.” not a tilde
-
References: The referencing style is inconsistent and does not follow standard citation formats. All in-text citations and reference list entries should be reviewed and corrected accordingly.
-
Overstated Claims: Some statements are unnecessarily pretentious and lack sufficient scientific backing. For example:
“Chitinase 3-like 1 (CHI3L1/YKL-40/Brp-39) is a key molecule involved in inflammation-12 associated cancer and serves as a crucial link...”
This sentence overstates the role of the molecule. Claims should be stated more cautiously.
This paper has merit and may contribute meaningfully to the field upon revision. However, the current version is not yet ready for publication due to the high number of technical, typographical, and stylistic issues. A careful review and correction of these elements is essential.
Comments on the Quality of English LanguageEnglish needs some polishing
Author Response
Reviewer 2:
She submitted manuscript addresses a relevant topic and presents findings that could contribute meaningfully to the field, congratulations to the authors! The manuscript demonstrates potential. However, in its current form, the manuscript requires significant technical and linguistic revision before it can be considered for publication.
Major concerns:
Comment 1) Language and Grammar:
The text contains numerous grammatical and syntactical errors that hinder readability and overall clarity. A thorough language edit by a native or fluent English speaker is strongly recommended.
Response-1) Thank you very much for pointing out grammatical and syntactical errors in this manuscript. We have revised them as much as possible, and asked for language edit by a native English speaker (Ms. Kori Aiken).
Comment-2) Formatting Issues:
The font is not uniform throughout the document, which disrupts the visual consistency.
- The title includes a full stop, which is not standard practice and should be removed.
- Several typographical errors are present (e.g., inconsistent spacing, punctuation misuse, etc.).
Response-2) Thank you for your valuable comments. We have changed our title as ““Future perspectives and conclusions from animal models of CHI3L1-related inflammation-associated cancer”. Typographical errors are corrected professionally by edit by a native English speaker (Ms. Kori Aiken).
Comment-3) Inaccurate Symbols and Units: In Table 2, the range should be replaced with a proper hyphen: “98–100.” not a tilde
Response-3) Thank you for your kind advice. Now we have edited the reference parts in Table 2.
Comment-4) References: The referencing style is inconsistent and does not follow standard citation formats. All in-text citations and reference list entries should be reviewed and corrected accordingly.
Response-4) Thank you for your advice. Now we have reviewed and corrected the reference section accordingly.
Comment-5) Overstated Claims: Some statements are unnecessarily pretentious and lack sufficient scientific backing. For example:
“Chitinase 3-like 1 (CHI3L1/YKL-40/Brp-39) is a key molecule involved in inflammation-12 associated cancer and serves as a crucial link...”
This sentence overstates the role of the molecule. Claims should be stated more cautiously.
Response-5) Thank you for your comment. We have revised the sentence to avoid overstatement and better reflect on the current evidence. The revised sentence now reads: “Chitinase 3-like 1 (CHI3L1/YKL-40/ Brp-39) is increasingly recognized as an important mediator in inflammation-associated cancer, with evidence suggesting its involvement in the interplay between chronic inflammation and epithelial tumorigenesis”.